# Species-specific quantification of circulating ebolavirus burden using VP40-derived peptide variants

**Qingbo Shu**[1], **Tara Kenny**[2], **Jia Fan**[1], **Christopher J. Lyon**[1], **Lisa H. Cazares**[2], **Tony Y. Hu**[1]*

**1** Center for Cellular and Molecular Diagnostics, Department of Biochemistry and Molecular Biology, School of Medicine, Tulane University, New Orleans, Louisiana, United States of America, **2** Systems and Structural Biology Division, Protein Sciences Branch, U.S. Army Medical Research Institute of Infectious Diseases, Frederick, Maryland, United States of America

* tonyhu@tulane.edu

**Data Availability Statement:** The authors confirm that all data underlying the findings are fully available without restriction. The mass spectrometry data have been deposited to the Panorama Public with the link https://

## Abstract

Six ebolavirus species are reported to date, including human pathogens Bundibugyo virus (BDBV), Ebola virus (EBOV), Sudan virus (SUDV), and Taï Forest virus (TAFV); non-human pathogen Reston virus (RESTV); and the plausible Bombali virus (BOMV). Since there are differences in the disease severity caused by different species, species identification and viral burden quantification are critical for treating infected patients timely and effectively. Here we developed an immunoprecipitation-coupled mass spectrometry (IP-MS) assay for VP40 antigen detection and quantification. We carefully selected two regions of VP40, designated as peptide 8 and peptide12 from the protein sequence that showed minor variations among Ebolavirus species through MS analysis of tryptic peptides and antigenicity prediction based on available bioinformatic tools, and generated high-quality capture antibodies pan-specific for these variant peptides. We applied this assay to human plasma spiked with recombinant VP40 protein from EBOV, SUDV, and BDBV and virus-like particles (VLP), as well as EBOV infected NHP plasma. Sequence substitutions between EBOV and SUDV, the two species with highest lethality, produced affinity variations of 2.6-fold for p8 and 19-fold for p12. The proposed IP-MS assay differentiates four of the six known EBV species in one assay, through a combination of p8 and p12 data. The IP-MS assay limit of detection (LOD) using multiple reaction monitoring (MRM) as signal readout was determined to be 28 ng/mL and 7 ng/mL for EBOV and SUDV respectively, equivalent to ~1.625–6.5×10$^5$ Geq/mL, and comparable to the LOD of lateral flow immunoassays currently used for Ebola surveillance. The two peptides of the IP-MS assay were also identified by their tandem MS spectra using a miniature MALDI-TOF MS instrument, greatly increasing the feasibility of high specificity assay in a decentralized laboratory.

## Author summary

We developed a quantitative assay for carefully selected species-specific VP40 peptide variants by selecting two VP40 tryptic peptides (i.e., LGPGIPDHPLR from EBOV VP40 and

panoramaweb.org/1A7WxR.url. They were also deposited to the ProteomeXchange Consortium [1] with the dataset identifiers PXD021149. The MALDI-TOF mass spectrometry data of LOD using recombinant VP40 haven been deposited to the ProteomeXchange Consortium with the dataset identifiers PXD026048. Reference: [1] Perez-Riverol Y, Csordas A, Bai J, Bernal-Llinares M, Hewapathirana S, Kundu DJ, et al. The PRIDE database and related tools and resources in 2019: improving support for quantification data. Nucleic Acids Res. 2019;47(D1):D442-D50.

**Funding:** The work was primarily supported by research funding provided by National Institutes of Health to T. H. (R01HD090927, R01HD103511, R01AI113725 and R21AI126361), The U.S. Department of Defense to T. H. (W8IXWH1910926), and Tulane University Weatherhead endowment fund to T. H. The funders had no role in study design, data collection and analysis, decision to publish, or preparation of the manuscript.

**Competing interests:** I have read the journal's policy and the authors of this manuscript have the following competing interests: T.Y.H. and C.J.L. report other interests from NanoPin Technologies, Inc., outside the submitted work. In addition, T.Y.H. and Q.S. has a patent ("Compositions and methods of determining a level of infection in a subject") licensed to NanoPin Technologies, Inc. The rest of us declare no competing interests.

LRPILLPGK from BDBV VP40) to generate antibodies pan-specific for variants of these peptides. Since species-specific structural difference may affect VP40 liberation during sample inactivation, and sequence variations may affect target peptide capture by pan-specific target antibodies, we quantified VP40 in EBOV and SUDV virus like particles (VLPs) to build species-specific standard curves to account for the greatest differences and improve the accuracy of VP40 quantitation. We also evaluated the feasibility of using miniature mass spectrometer to collect the tandem MS spectra for both target peptides.

## Introduction

Differences in the lethality of the six ebolavirus species has been well documented[1–3]. BDBV, EBOV and SUDV ebolavirus are highly lethal human pathogens. TAFV ebolavirus causes severe but nonlethal human disease. RESTV ebolavirus only caused asymptomatic human infection. The pathogenicity of BOMV ebolavirus for humans is unclear[4]. Currently there are two mAb-based drugs (Inmazeb and Ebanga) for ebolavirus disease (EVD) treatment approved by FDA[5], and they were proved to be efficient for EBOV based on a clinical trial [6]. The two drugs have not been evaluated for efficacy against species other than EBOV. Nonetheless, the difference in optimum dosage and timing of mAb treatment between EBOV and SUDV infection was reported in mouse models[7,8]. This indicates the importance of species identification during an outbreak. Initial viremia is associated with outcome of the individual and outbreak duration[9]. Viral load monitoring may assist in risk stratification of EVD patients[3]. As a result, quantitative detection of viral burden is an important consideration for determining therapeutic options[10].

Current WHO recommended tests for EVD diagnosis include automated or semi-automated nucleic acid tests (NAT), including RT-PCR, and rapid antigen detection tests[11]. Though NAT assays are widely used, factors such as cost, time to processing (including sample storage temperature constraints and transport time), availability, and required level of operator expertise contributed to delays in provision of rapid results during the 2013–2016 Western African EVD outbreak[3]. Rapid, simple, sensitive, and safe EBOV diagnostic tests are still needed[12]. The performance of NAT assays has been found to vary even among national reference laboratories. False-negative results are not uncommon due to PCR inhibitors present across specimen types, as well as sequence variation in novel virus strains/species[13]. Appropriate sample handling prior to analysis to avoid RNA degradation and avoidance of cross-contamination are required to generate reliable results. Another limitation of the available PCR assays is their limited species coverage[14]. Of the ten assays currently used in the field, only seven detect EBOV; one detects EBOV and SUDV, and two detect all species without species differentiation[3,15]. A reverse transcription loop-mediated isothermal amplification (RT-LAMP) assay can achieve species identification within 15 minutes after sample inactivation and RNA extraction[16,17], but is unable to assess viral load accurately. Nanopore sequencing was recently applied to sequence the ebolavirus genomes in a real-time fashion, which enabled identifying species and mutations[18], but the ability of quantitative assessment of viral load by nanopore sequencing has yet to be validated.

Quantitative detection of viral antigens is an alternative way in this regard. Ebolavirus antigens concentrations in blood typically reach high levels within the first days of symptoms[19], and they are more stable than RNA during ebolavirus inactivation and sample transportation. The ebolavirus protein VP40 is the most abundant viral protein[20]. Therefore, VP40 has been selected as a diagnostic marker in the three commercially available rapid diagnostic tests

(RDTs) that have received WHO and/or FDA EUA status[13]. However, none of these RDTs has the ability to differentiate virus species and estimate viral burden due to their scaled-down design. A quantitative method that can distinguish among ebolavirus species would thus be highly useful to aid the precise diagnosis and personalized management of EVD.

Species-specific peptide variants of ebolavirus antigen VP40 could be promising biomarkers. Unlike traditional immunoassays employed in RDTs, identification of VP40 peptides by MS enables species differentiation based on sequence substitutions, although VP40 sequences are highly conserved among ebolavirus species. Furthermore, it can provide the quantitative information of VP40 that may reflect viral burden in circulation. Immunoassays detecting full-length VP40 protein have been established, but immunoassays targeting species-distinguishable peptide variants of VP40 have not been explored. An ideal VP40 peptide-based immunoassay would require careful selection of peptide targets to reach the goals of species identification and VP40 quantification. To achieve the needed sensitivity of a blood-based test in identifying EVD patients at their early phase of infection, high-affinity antibodies are required for the efficient immunoprecipitation of selected target peptides.

Here we successfully selected two VP40 regions based on the sequence information and MS analysis results of recombinant protein digests. The two peptides showed minor sequence substitutions that enable species differentiation. The substitutions also cause affinity variation of the peptide variants against their pan-variants antibodies; therefore, we evaluated the affinity differences first and validated the feasibility of capturing all variants using samples containing VP40 from EBOV, SUDV and BDBV. We also explored the use of a mini MALDI device as a solution for low resource field laboratories. The ability to generate high-resolution tandem MS scan of VP40 peptide targets in this device greatly improved assay specificity and throughput.

## Results

### LC-MS analysis of EBOV and SUDV VP40 digests to identify candidate biomarker peptides for antibody production

Since VP40 is the most abundant component of ebolaviruses and thus a good biomarker candidate, trypsin digests of recombinant EBOV and SUDV VP40 protein were analyzed by LC-MS to identify candidate VP40 peptides that can distinguish EBOV and SUDV. Detected peptides were mapped to VP40 sequence using MaxQuant software default parameters ($\geq 7$ amino acids and $< 4600$ Da). This analysis identified 15 overlapping peptides (p1, p3-p10, p12-p17) that exhibited variable conservation (**Fig 1A–1C**). A predicted C-terminal peptide was not identified in the LC-MS data of either digest, likely due to inefficient ionization since its single cleavage site was efficiently cleaved to produce p17 in both VP40 digests. Sequence variation also resulted in alternate cleavage sites for the second linear peptide of these proteins, while differential cleavage at an unfavorable cleavage site (arginine followed by a proline) resulted in detection of p11 in the trypsin digest of the SUDV but not EBOV recombinant protein. Both these peptides were therefore excluded from further evaluation.

Four VP40 peptides (p3-p4 and p10-p11, **Fig 1B**) were also excluded as potential biomarker candidates since their generation required cleavage at an unfavorable trypsin site (arginine before proline) that can be cleaved with variable efficiency[21,22], as revealed by the absence of p11 in the EBOV digest (**Fig 1A**). Six peptides were excluded for lacking seven sequential amino acids without a nonconservative substitution (p1, p3, p7, p13-p15), due to the fact that peptide-specific antibodies are reported to recognize epitopes containing 7 to 9 contiguous peptides, and 85% of epitopes with linear components recognize five or more contiguous amino acids[23]. Finally, eight VP40 peptides were excluded for containing one or more oxidation-prone amino acid residues (methionine, cysteine, or tryptophan: peptides p1, p3-p6,

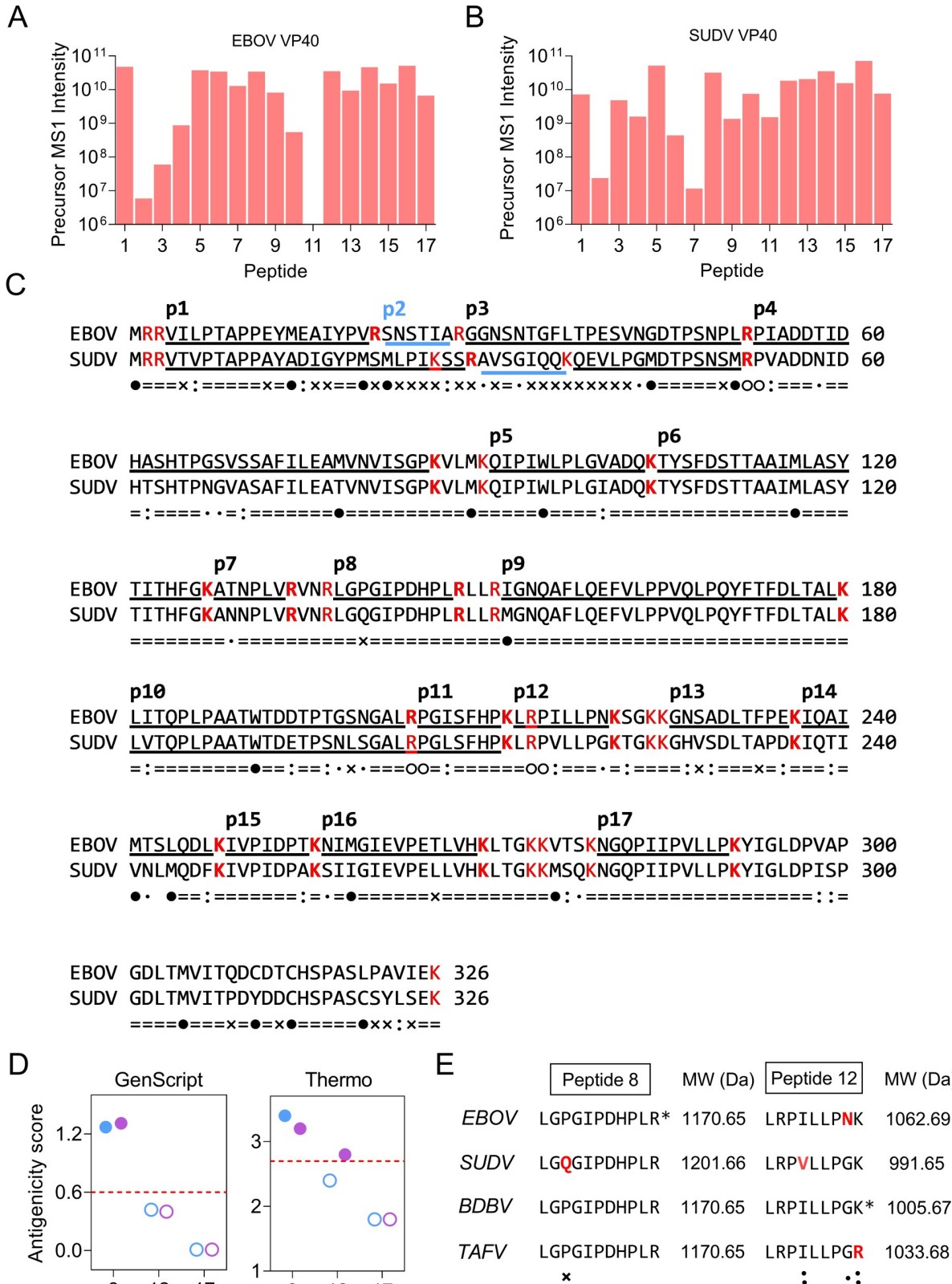

**Fig 1. Selection of Ebolavirus peptides for species identification.** (A-B) The tryptic peptides identified via LC-MS/MS derived from (A) EBOV and (B) SUDV recombinant VP40 protein. The precursor MS1 intensities are calculated in Maxquant software as the summed up

extracted ion current of all isotopic clusters associated with the identified peptide sequence. Peptides are numbered according to their order in the linear sequences. (C) Map of EBOV VP40 tryptic peptides ≥ 7 amino acids in length. Peptide 2 (p2, blue underlines) is not aligned between the two species. The labels below indicate amino acid residues that are prone to be oxidized (•); identical between EBOV and SUDV VP40 (=); non-conservative substitution (x); conserved substitution between groups of strongly similar properties, i.e., scoring > 0.5 in the Gonnet PAM 250 matrix (:); conserved substitution between groups of weakly similar properties, i.e., scoring = < 0.5 in the Gonnet PAM 250 matrix (.); and arginine or lysine followed by proline motif that are prone to be miss cleaved by trypsin (○○). Red residues indicate potential tryptic cleavage sites. The peptides are numbered identically as shown in (A-B). (D) Candidate peptide antigenicity determined by predictive algorithms from (left) Genscript or (right) Thermo Scientific. Blue and purple dots respectively indicate data for the EBOV and SUDV species, while the closed or open dots indicate the antigenicity scores for a given peptide sequence that are above (closed) or below (open) the antigenicity score threshold (dotted red line) of a good antigen peptide defined by each algorithm. (E) Sequence alignment and molecular weights of VP40 peptide 8 and 12 regions for the four species known to causing human infections. Conservative (: or.) and divergent (x) substitutions are marked in red. Asterisks indicate the sequences serving as antigens to generate peptide antibodies.

p9, p14, and p16), since oxidation of these residues could inhibit formation and stability of the corresponding antibody-peptide complexes and complicate their specific detection by LC-MS analysis[22].

Analysis of the antigenic potential of the three remaining peptide candidates (p8, p12, and p17) by two antigenicity scoring tools (GenScript's OptimumAntigen Design Tool and Thermo Scientific's Antigen Profiler Peptide Tool) was performed to identify the most likely candidates for specific antibody production. This analysis found that only p8 and p12 approached or exceeded the threshold for desirable antigenicity using either scoring method (Fig 1C and 1D). The sequence of p17 showed species-specificity in BOMV only (S1 Fig), suggesting that it could be used to identify this plausible ebolavirus species if its peptide specific antibody is generated in the future. In the current work, we selected p8 and p12 to distinguish the most possible species.

## Enrichment of VP40 peptide variants using pan-peptide variants antibody

Both p8 and p12 could distinguish VP40 expressed by EBOV and SUDV, the two ebolaviruses that have been responsible for most EVD outbreaks to date, but exhibited differential ability to distinguish other ebolaviruses, including other ebolaviruses that causes severe EVD, BDBV and TAFV (Fig 1E). VP40 p8 was conserved among all ebolaviruses known to infect humans except EBOV, where it differed by single amino acid substitution. VP40 p12 variants differed at one internal and two adjacent C-terminal sites among these species, although two of these differences entailed conservative substitutions. These species-specific p12 differences could potentially identify three of the four ebolaviruses that cause severe EVD in humans and which have been responsible for all symptomatic ebolavirus cases to date.

Sequence differences among these peptides could also influence their capture by specific antibodies. Antibodies generated against VP40 p8 should demonstrate equal affinity with all four target ebolavirus species expect perhaps SUDV, which contains a single non-conservative amino acid substitution, allowing p8 to serve as a general marker of ebolavirus infection. VP40 p12 sequence variation could permit the specific identification of all four species, but could reduce the affinity of p12 capture antibodies for species-specific p12 peptides by attenuating or disrupting critical amino acid interactions involved in the formation of the antibody-peptide complexes. The simple assay design would be capturing all peptide variants using single peptide antibody. To reduce the effect of sequence variation, we therefore generated peptide-specific antibodies to p8 and p12 and evaluated their ability to detect p8 and p12 IP-MS signal from trypsin digests of normal plasma spiked with virus-like particles (VLPs) that expressed EBOV or SUDV VP40. For this analysis, VP40 p8 specific antibodies were generated using a consensus sequence that differed only for SUDV, while p12-specific antibodies were produced using the BDBV p12 sequence, which differed by a single amino acid from p12 peptides produced by the three other species that cause human EVD (Fig 1E).

## Design of IP-MS assay for VP40 peptide variants

There are five major priorities for an MS-based VP40 assay, including (1) rapid and safe sample processing, (2) a scalable and portable signal readout, (3) low cost, (4) high accuracy, and (5) species differentiation. We selected the CDC recommended sample inactivation method, which used Triton X-100 to disrupt the intact ebolavirus particles and release VP40 protein (**Fig 2A**), as a starting point since Triton X-100 in a low concentration is MS compatible. Briefly, the samples were incubated with 0.5% Triton X-100 and heated at 100˚C for 5 minutes to denature plasma proteins (See **Methods** for details). This inactivation approach avoided the need for buffer exchange and permitted all sample processing steps after inactivation to be conducted in a BSL-2 laboratory. To make this workflow scalable, we avoided depletion of high-abundance plasma proteins, enriched the VP40 tryptic peptides directly from plasma digest, and eliminated protein reduction and alkylation steps commonly used in proteomics studies to reduce processing time. This workflow should also be readily automated using an established automated magnetic beads immunoprecipitation platform[24]. The entire assay workflow required about 3.5 hours, making it suitable for screening a large population during an outbreak.

After optimization of inactivation, digestion and IP of peptides, the proposed IP-MS assay of VP40 peptide variants showed a similar extend of ease of sample pre-treatment and a similar

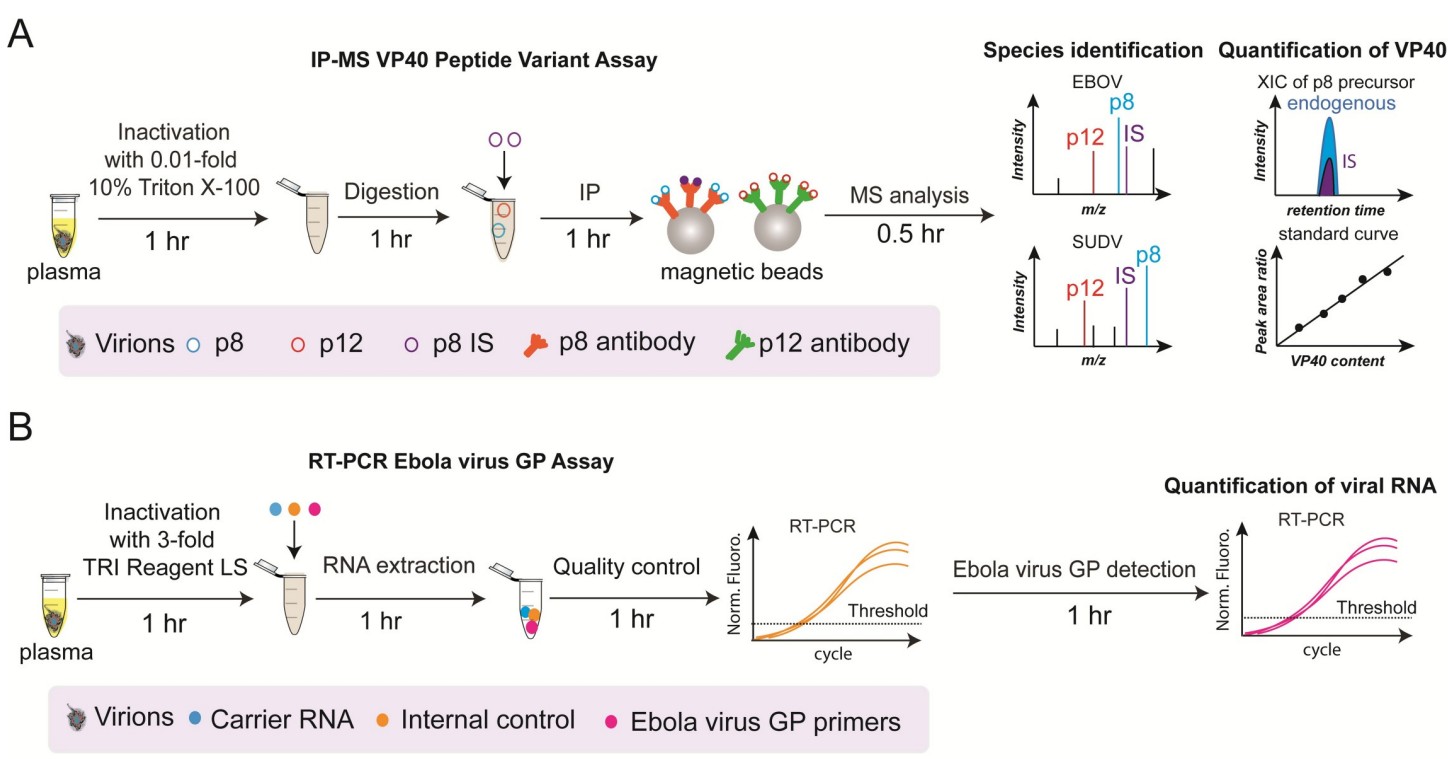

**Fig 2. Comparison of the IP-MS assay for VP40 peptide variants with an RT-PCR assay for the GP gene.** Both assays employ four steps for sample analysis. (**A**) In the IP-MS assay for VP40 peptide variants, samples are inactivated using a method described by 2005 CDC Interim Guidance and digested to release the target VP40 peptides, which are then immunoprecipitated, and analyzed by mass spectrometry to detect mass differences among the peptide 8 and 12 sequences encoded by different ebolavirus species. The VP40 p8 and p12 target peptides and internal standard (IS) peptide are captured during immunoprecipitation (IP) by target-specific antibody-coupled magnetic beads. The peak area ratio between known amounts of VP40 p8 (blue peak) and its corresponding IS peptide (purple peak) is used to generate a standard curve to permit quantification of endogenous VP40 in clinical samples. (**B**) In the RT-PCR assay for the GP gene, a carrier RNA and a high concentration internal control are spiked into the sample before extraction. Each extracted RNA sample is tested with the internal control RT-PCR RNA Assay to evaluate the yield of the spiked-in internal control. If the internal control amplified within manufacturer-designated ranges, further quantitative analysis of the viral target is performed using standard procedures.

3–4 hours turnaround time when compared to the widely used RT-PCR assay targeting ebolavirus GP gene (**Fig 2B**). VP40 antigen level serves as a critical parameter in monitoring the disease progression and prognosis. By developing the accurate quantification method for VP40, it is possible to study the correlated changes of plasma VP40 level with the viral load quantified by the RT-PCR assay. Notably, amino acid mutations provide the species distinguishable ability of this IP-MS assay, whereas it would affect the antibody binding affinity, therefore, we evaluate this effect first.

To evaluate the affinity difference of p12 variants against the single peptide antibody we generated, a multiple reaction monitoring (MRM) assay was developed. Six MRM transitions were selected for both EBOV and SUDV VP40 p12 based on their tandem MS spectra to specifically identify the two variants (**Fig 3A** and **3B**). Recombinant VP40-spiked PBS dilutions were tested using the MRM assay targeting p12. Expected co-elution of the six transitions were observed for both peptide variants at 16.2 and 15.8 min respectively, indicating good chromatographic separation (**Fig 3C** and **3D**). The summed chromatographic peak areas were 10-fold higher in PBS samples spiked with similar amounts of SUDV VP40 when compared to EBOV VP40, as evaluated by the y-intercept of standardized dilution curves of these samples (**Fig 3E** and **3F**). The observed difference of peak area could be explained by a combined effect of efficiency variations due to trypsin digestion, antibody immunoaffinity enrichment and LC-MS chromatography and ionization. LC-MS analysis of recombinant SUDV vs. EBOV VP40 trypsin digests detected a 0.52-fold difference between EBOV and SUDV peptide 12 precursor MS1 intensities ($1.84 \times 10^{10}$ vs $3.52 \times 10^{10}$; **Fig 1A** and **1B**), thus reflecting the effects of variable trypsin digestion and MS ionization. Therefore, since the digestion and ionization efficiency differed by 0.52-fold, and the relative fold difference in assay response differed by 10-fold between the two species, EBOV and SUDV, the affinity difference can be inferred from the two values as 19-fold (10/0.52). Taken together, these results suggest that the two p12 variants showed a 19-fold difference in their affinities caused by the less conserved N>G substitution in the penultimate position of SUDV VP40 (**Fig 1E**).

## Quantification of VP40 from VLP-spiked plasma by IP-LC-MS analysis of p8 and p12 signal

Plasma samples were spiked with non-infectious virus-like particles (VLPs) containing EBOV or SUDV VP40 and glycoprotein (GP) (**Fig 4A**)[25], to mimic IP-MS analysis of ebolavirus biomarkers peptides from digested plasma of individuals with ebolavirus infections. VP40 p8 and p12 signal was detected using the multiplexed MRM method developed using transition ions selected from the LC-MS/MS spectra of p8 and p12 from trypsin digests of recombinant EBOV and SUDV VP40.

EBOV and SUDV VLP protein extracts were estimated to contain similar VP40 content (22.4 ± 1.3% vs. 28.0 ± 3.3%) when evaluated by a densitometric analysis comparing western blot VP40 signal in VLP protein extract dilutions versus standard curves of known amounts of recombinant VP40 (**Fig 4B**). Similar results were obtained when the VP40 content of EBOV and SUDV VLPs was measured against IP-MS assay standard curves generated for EBOV or SUDV recombinant VP40 protein digests (18.8 ± 3.5% vs. 21.0 ± 2.6%; **S1 Table**). No significant difference in VLP VP40 content was detected by either method.

Comparison of VP40 p8 and p12 MRM signal from healthy-donor plasma samples spiked with EBOV and SUDV VLP samples containing approximately equimolar amounts of VP40, revealed species-specific peak retention times and extracted MRM transition ion chromatograms that readily distinguished their species of origin, as well as signal intensity differences (**Fig 4C** and **4D**). VP40 p8 peak areas were 2.6-fold higher in healthy plasma samples spiked

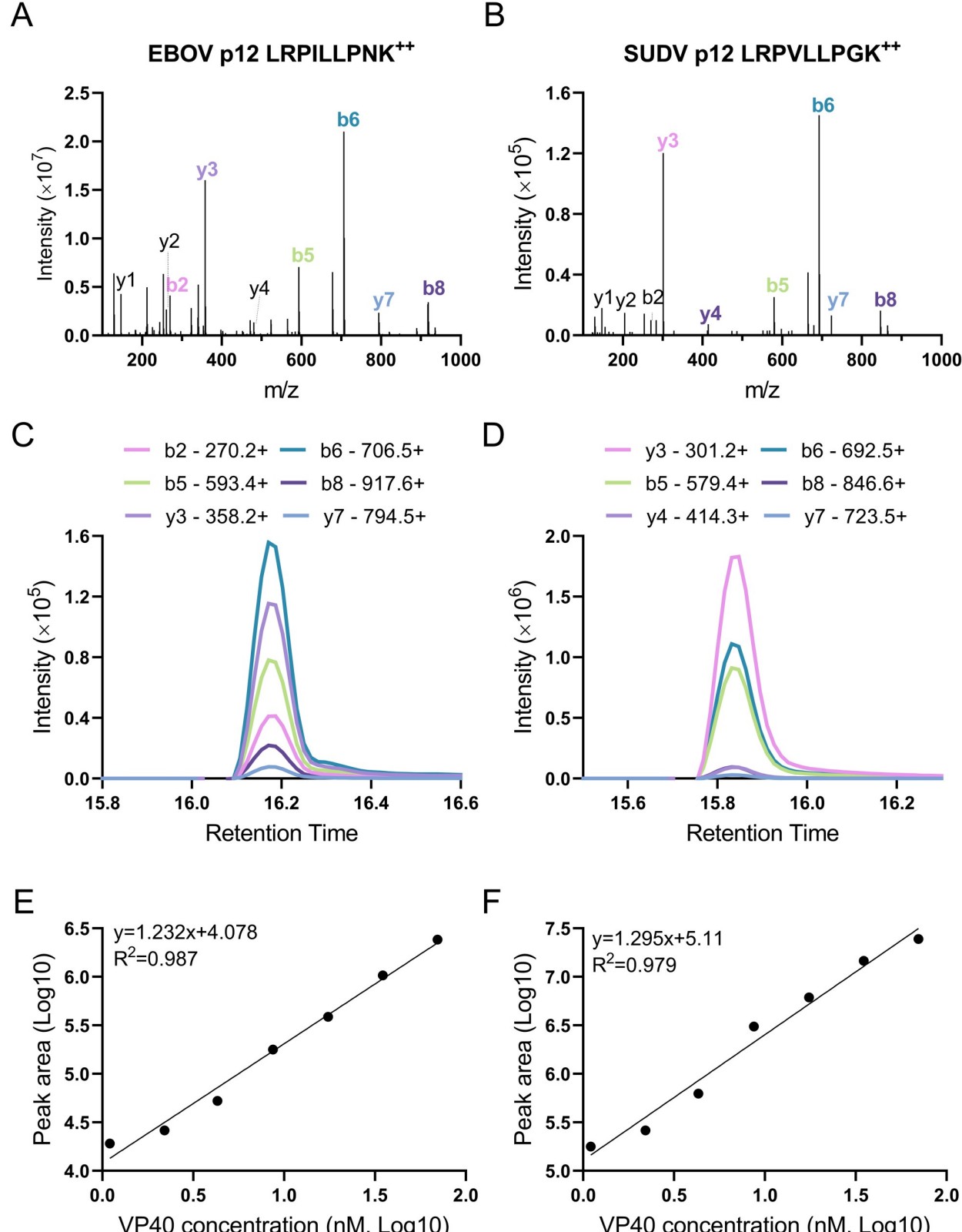

**Fig 3. Antibody affinity differences against two VP40 p12 variants measured by MRM.** (A-B) The tandem MS spectra of (A) EBOV and (B) SUDV VP40 p12 identified in recombinant protein tryptic digests. All matched ion peaks are annotated, with the six selected p12 MRM transitions

colored. (C-D) Extracted ion chromatograms of six MRM transitions in (C) EBOV and (D) SUDV recombinant VP40 spiked-in PBS samples at a concentration of 70 nM. (E-F) Curve fitting between p12 MRM transition summed peak area and VP40 concentration in (E) EBOV and (F) SUDV VP40 spiked-in PBS samples.

with dilutions of SUDV vs. EBOV VLPs as determined by comparison of the slopes of their standardized dilution curves (**Fig 4E**), indicating differences in the affinity enrichment and/or separation/ionization of these sequence variants in plasma. LC-MS analysis of recombinant SUDV vs. EBOV VP40 protein digests, however, differed by <10% ($3.18 \times 10^{10}$ vs $3.42 \times 10^{10}$; **Fig 1A** and **1B**), implying these peptides had similar separation/ionization characteristics, and thus likely differed in their affinity enrichment in plasma due to a single non-conservative amino acid substitution in the SUDV p8 sequence.

Given the lower variability of VP40 p8 signal in comparison with p12, standard curves were generated by spiking healthy plasma with serial dilutions of EBOV and SUDV VLPs to determine the linearity and limit of detection (LOD) for EBOV and SUDV VP40. In these analyses, samples were spiked with a constant amount (2.5 pmol) of synthetic heavy-isotope-labeled p8 internal standard peptide after trypsin digestion and the ratio of p8 VP40 to internal standard signal was plotted to normalize IP-MS signal and to estimate VP40 content for any signal value within the linear range. Samples spiked with EBOV or SUDV VLPs demonstrated similar linearities but different slopes and LOD that reflected the greater affinity of the capture antibody for the SUDV p8 sequence (**Fig 4E and 4F**). SUDV and EBOV VLP LODs were detected in samples containing 2.5 ng and 12.5 ng of total VLP protein and estimated to contain 0.7 and 2.8 ng of VP40, and both demonstrated good linearity ($R^2 > 0.98$) of quantitation over a broad range of VLP concentrations.

To further validate the IP-LC-MS method in other less lethal ebolavirus species, i.e., BDBV, TAFV and RESTV, we selected six MRM transitions for each peptide variants from their VP40. The p8 and p12 peptide variants were detected in a reproducible way in the tryptic digest of the three species' virions inactivated by irradiation (**S3** and **S4 Figs**). The MS signal of two peptides were also observed in human plasma spiked with the inactivated virions, and the p8 peptide showed a linearity of 0.984–0.987 (Pearson correlation R square) in the viral protein concentration range of 0.625–10 μg/mL (**Fig 5**). As a result, our assay enables safe and rapid ebolavirus species identification and VP40 quantification.

## Quantification of VP40 in NHP plasma by IP-MALDI-MS

MS-based applications employed in research and clinical laboratories primarily employ LC-MS or matrix-assisted laser desorption/ionization time-of-flight (MALDI-TOF) MS for target identification and quantification, with each approach demonstrating strengths and weaknesses. MALDI-TOF MS assays permit high-throughput, while LC-MS applications, which typically employ electrospray ionization (ESI) offer greater analytical sensitivity and specificity although detection performance can also be influenced by the differential ionization efficiency of MALDI or ESI for target peptides. Both ESI and MALDI ion sources are employed in miniature MS instruments that may be useful for rapid diagnosis in future EVD outbreaks.

To validate the feasibility of using MALDI-TOF MS in the proposed assay, we employed a Bruker Microflex LRF benchtop MALDI-TOF MS to detect the two peptide targets, p8 and p12. Considering the inherited ionization difference between ESI and MALDI sources, we first examined the recombinant protein digests of VP40. MALDI-TOF MS analysis of EBOV and SUDV recombinant VP40 protein digests detected six and nine of the 17 tryptic peptides detected by LC-MS within a precursor mass error of 1 Da, and both the p8 and p12 biomarker

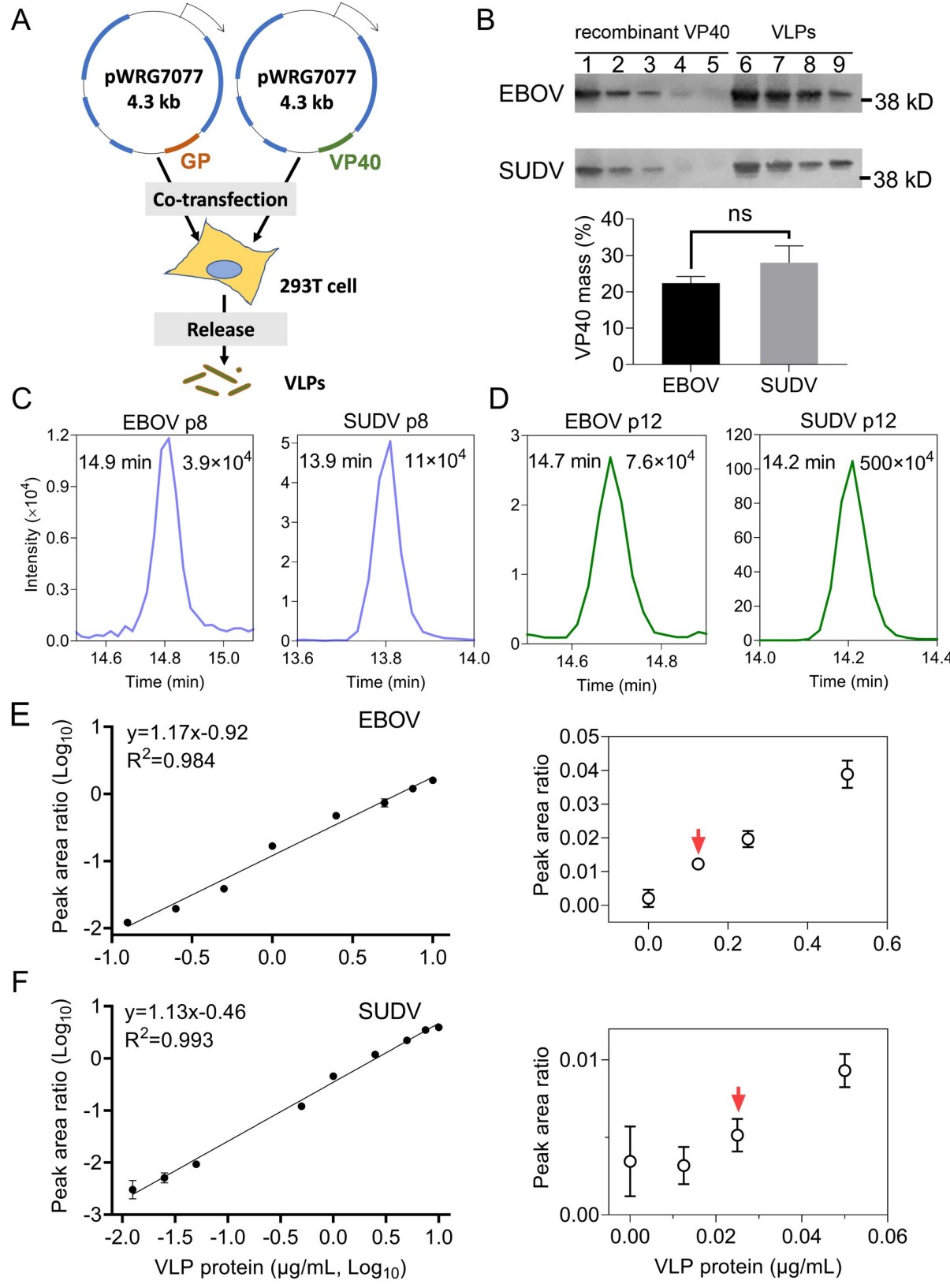

**Fig 4. Species-specific VLP used for VP40 quantification.** (A) Generation of VLPs by co-transfecting 293T cells with pWRG7077 plasmid containing VP40 (orange) and the plasmid containing GP genes (green) from ebolavirus. (B) Western blot quantification of VP40 content in

VLPs. Wells 1–5 and 6–10 respectively indicate two-fold serial dilutions of recombinant VP40 protein (starting from 0.87 μg of VP40) and VLP protein extract (starting from 2.5 μg total VLP protein). The position of a 38 kDa protein marker that migrates close to VP40 is indicated next to each Western blot. Data is shown as mean with standard deviation in the bar plot (n = 2). (C-D) Extracted ion chromatograms of VP40 peptide 8 (C) and peptide 12 (D) in human plasma spiked with 50 ng of EBOV or SUDV virus-like particles expressing VP40. The measured retention time and peak area of each peptide are shown in the left and right upper corners respectively in the upper panels. (E-F) Quantification curve for (E) EBOV and (F) SUDV VP40 peptide 8 in human plasma spiked with the corresponding VLPs. The extracted peptide peaks were used to calculate the peak area ratio for (left) the standard curve of VP40 peptide signal to VLP protein input, and (right) the LOD (red arrow). Data is shown as mean with standard deviation (n = 3).

candidates were detected with strong signal intensity in each of these samples (S4 Fig). Species-selective variants of both peptides were also detected by MALDI-TOF-MS upon IP-MS analysis of plasma samples spiked with EBOV, SUDV or BDBV recombinant VP40 protein (S5 Fig). They were identified with less than 25 ppm mass error in both 10 μg VLP-spiked and authentic EBOV infected (Day 7, viral titer $7.91 \times 10^9$ Geq/mL) non-human primates (NHPs) plasma (Fig 6A). Due to the existence of lithium dodecyl sulfate (LDS) in the NuPAGE LDS sample buffer used for viral inactivation, the archived infected plasma sample at day 7 was processed first through filter-aided sample preparation protocol (See Methods for details). This helped to reduce the signal suppression effect of LDS on MALDI MS spectra, as further validated by VLP-spiked NHP plasma (Fig 6B). Nearly 10-fold difference in the slope indicated suppression of peptide signals with LDS buffer (Fig 6C), which resulted in 3.9-fold (240 vs 930 ng/mL) difference in LOD with or without LDS buffer during sample inactivation.

Longitudinal plasma samples were collected from two non-human primates (NHPs) at days 4 to 7 after their infection with EBOV, and analyzed by RT-PCR and Western blot to determine the correlation between circulating viral load and VP40 concentration (S6 Fig). Plasma EBOV viral loads detected in these animals at day 4 post-infection ranged from $2.9 \times 10^6$ for NHP 7284 and $1.2 \times 10^9$ Geq /mL for NHP 436. At day 6 post-infection, EBOV viral titer increased to $5.5–7.3 \times 10^9$ Geq /mL, consistent with the viral load detected in a previous NHP study[26].

Plasma levels of VP40 protein in the EBOV-infected NHPs were estimated to range from 226 to 847 μg/mL at day 6 post-infection when evaluated by a western blot densitometric analysis in comparison to a standard curve of recombinant EBOV VP40 protein (Fig 7). This quantitative analysis reached its LOD in the NHP 7284 plasma at day 5 post-infection, which showed a band with OD value lower than the lowest concentration standard on the same membrane. Therefore, the VP40 quantity cannot be estimated. When tested by IP-MALDI-MS, both p8 and p12 peptides were identified in the NHP 7284 plasma at day 6 post-infection, though the normalized p8 peptide peak intensity was low and we were not able to estimate the quantity of VP40 by the corresponding standard curve of VLP-spiked plasma. The same animal showed negative result for both peptides in the sample at day 5 post-infection. This can be explained by the profound influence of LDS buffer.

Extrapolation from the plasma VP40 concentration and EBOV load of these samples generated a linear equation with a slope as 23,130 (95% CI: 12,016–34,245, S6 Fig). The EBOV VP40 IP-MS LODs determined for LC-MS (28 ng/mL) and MALDI-TOF-MS (110 ng/mL; S1 Data), suggests that the assay LODs for EBOV virions in plasma using LC-MS and MALDI-MS are $\sim 6.5 \times 10^5$ and $2.5 \times 10^6$ EBOV Geq/mL. This estimate was performed to determine the potential LOD of the assay without the presence of LDS sample buffer, which is close to the mean EBOV levels in patient serum ($10^5$ to $10^6$ Geq/mL) at initial evaluation[9].

The major drawback of applying MS in diagnosing EVD is the requirement of high vacuum environment made by a powerful pumping system. To equip a MS in the new generation of mobile laboratories[27], miniature mass spectrometers are a promising choice that was

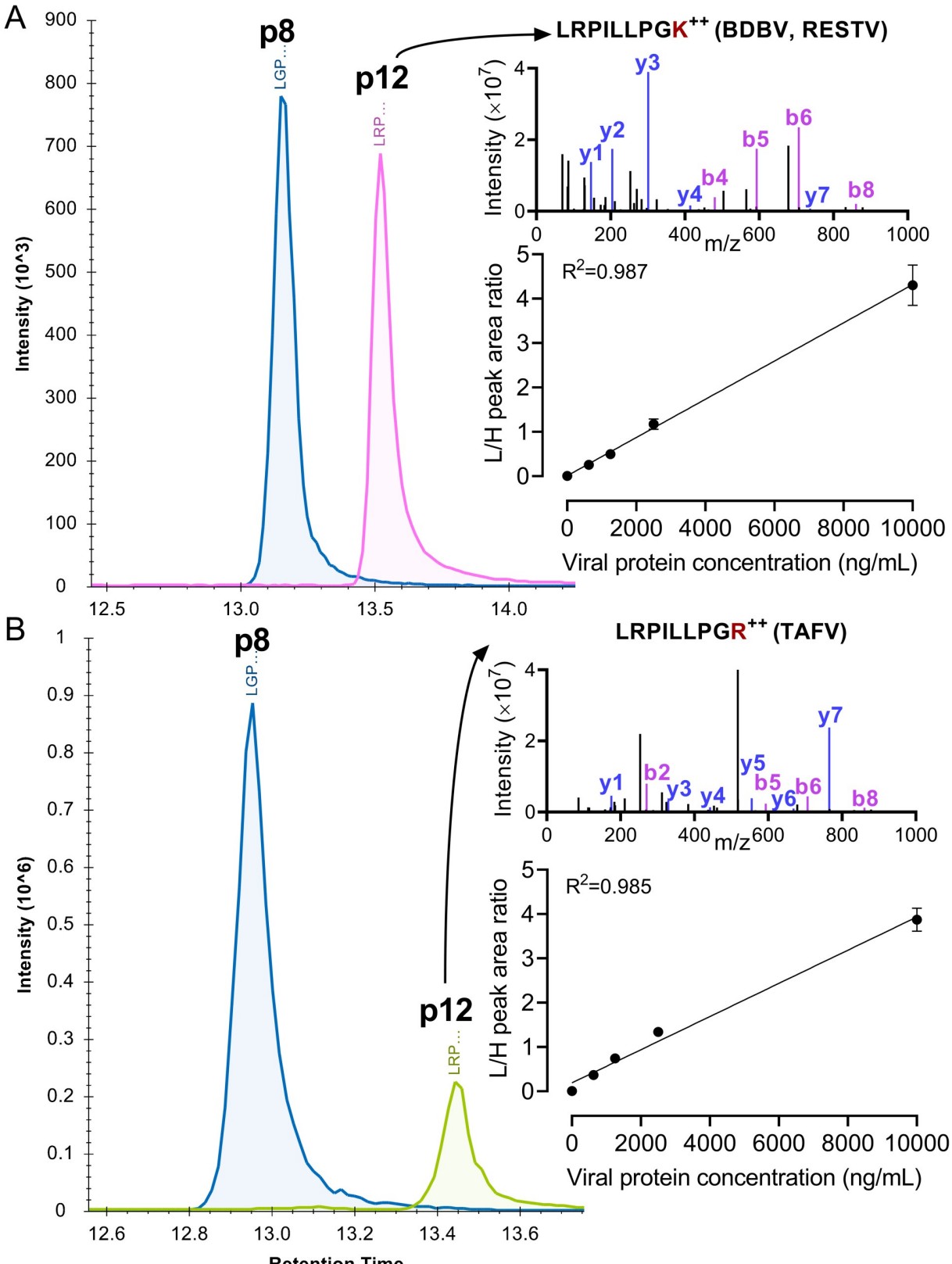

**Fig 5. Method validation using inactivated virions from less lethal species.** Extracted ion chromatograms of peptide 8 (blue line) and 12 (magenta and green lines) in the tryptic digest of (A) BDBV and (B) TAFV ebolavirus inactivated virions. The p8 sequence is shared by the

three species, whereas the p12 sequence differs by one amino acid located at the last position (K/R, red). Tandem MS spectra of p12 is presented on the upper right corner, and the major matched peptide fragmentation ions from high-resolution tandem MS spectra collection are annotated and colored (blue for y ions, purple for b ions). Below the spectra are standard curves generated by plotting p8 light and heavy version peak area ratio with the concentration of total viral proteins spiked into the sample. The linear correlation coefficients are indicated above the curves. Data is presented as mean±SD (n = 3).

recently applied to detect lipid species in tissue samples[28]. Representative commercial miniature MS instrumentation includes PurSpec miniβ, BaySpec Continuity and Shimadzu MALDI-mini-1. Although they differ in ion source design, all provide tandem MS analysis. Here we integrated peptide immunoprecipitation with the Shimadzu MALDImini-1. The streamlined configuration and lightweight design of the compact MALDImini-1 enable it to be more easily installed in a field or mobile laboratory[29]. To prove the feasibility of using miniature MS as the readout in this assay, we analyzed the peptide eluent from 10 μg/mL VLP-spiked human plasma using MALDImini-1. Both peptide 12 and peptide 8 were identified in the survey scan of peptide eluent with mass resolutions of 1,800–2,800 (**Fig 8**), suggesting that the two targets can be unambiguously identified with a high resolution using miniature MS. Through a tandem MS scan, b/y ions of peptide 8 were further identified with mass error< 0.6 Da. As a result, accurate precursor and product ion masses of the two peptide targets could be obtained in the miniature MALDI-TOF MS instrument.

## Discussion

Adequate viral inactivation methods were developed to avoid the need of BSL-4 biocontainment for large-scale EVD diagnosis based on NAT assays. Presently, laboratory EBOV inactivation is accomplished by gamma irradiation[30], UV radiation[31], nanoemulsion[32], and photoinducible alkylating agents[33], but these methods are not applicable in outbreak situations or as bedside inactivation methods. Other inactivation methods, such as acetic acid, heat, AVL buffer, TRIzol or the combination of heat and Triton X-100[9], are more applicable in outbreak situations and are currently used in field laboratories. A rapid bedside inactivation method was developed recently for nucleic acid tests[34]. The Magna Pure lysis/binding (MPLB) buffer containing 30–50% (w/w) guanidine thiocyanate and 20–30% (w/w) Triton X-100, according to the manufacturer's material safety data sheet[35] was used in this method. The ratio of MPLB buffer to EDTA-blood was 1:1. Guanidine thiocyanate is a non-polymer chemical compound used as a general protein denaturant, being a chaotropic agent, although it is most commonly used as a nucleic acid protector in the extraction of DNA and RNA from cells. Based on its chemical composition, it is assumed to have minimal effect on MALDI-MS detection of peptides detected after IP procedure and elution. In this study, it was proved that diluted Trition X-100 solution didn't affect the enrichment of peptides. These methods have now become routine procedures for NAT assays, though there are occasional cases where EVD diagnostic tests can be conducted at BSL-3 biocontainment[13] or even BSL-2 laboratory in a hospital[36].

Here we proposed to use the combination of heat and Triton X-100 for sample inactivation and developed an IP-MS assay that is compatible with this inactivation method. Due to the limited availability of clinical specimens that were processed following this preferred inactivation method, the current study was unable to evaluate this assay using archived clinical samples. However, it is reasonable to compare the method LOD with that of one of the LFIs ReEBOV RDT that can detect VP40 antigen and was validated using blood samples and in a field study[37,38]. The LOD of ReEBOV RDT was determined to be 4.7 ng/test in serum and 9.4 ng/test in whole blood, equivalent to $3.0\times10^{5}$–$9.0\times10^{8}$ genomes/mL. In comparison, the

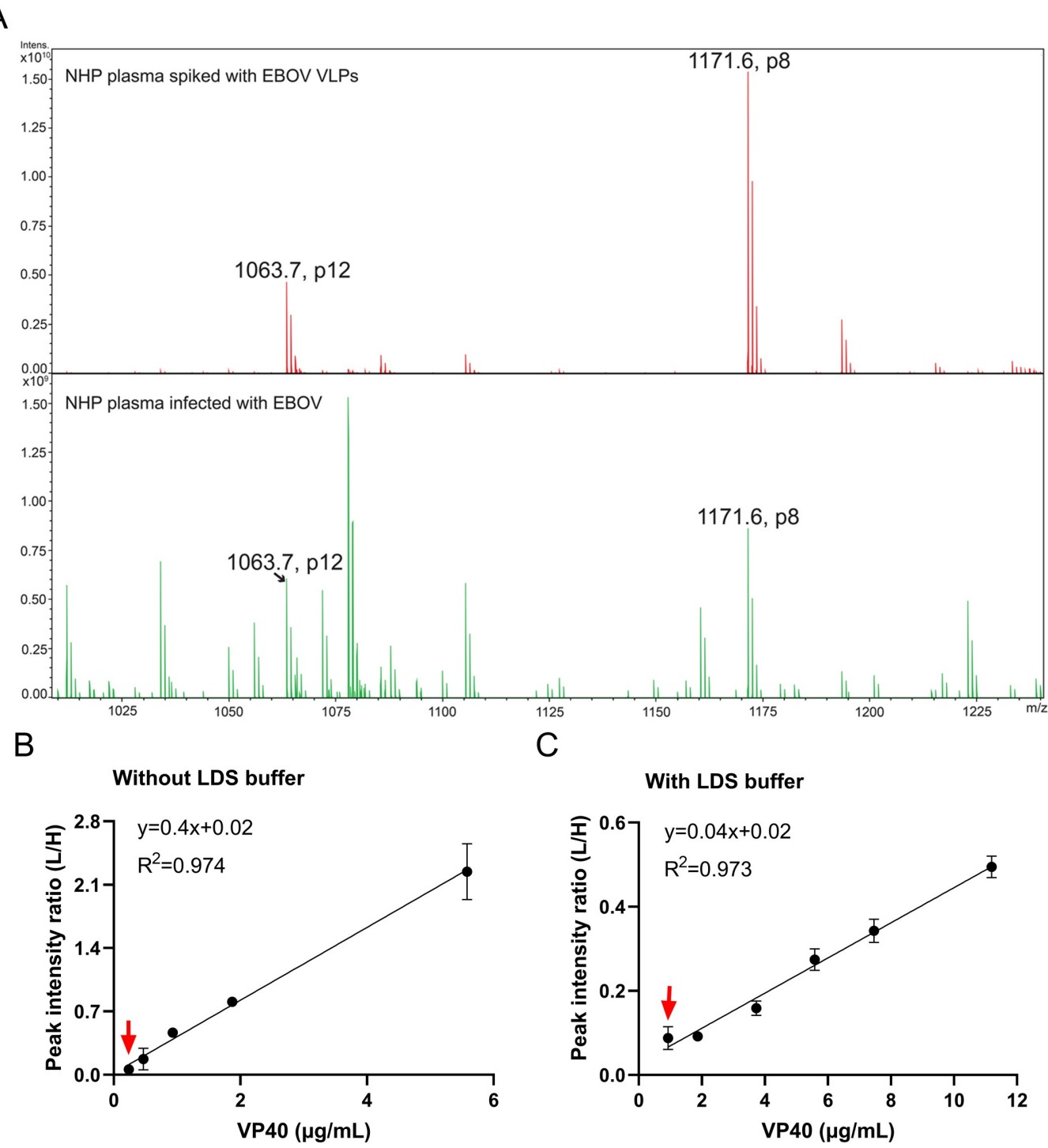

**Fig 6. Species identification and VP40 quantitation in NHP plasma by IP-MALDI-MS.** (A) MALDI-TOF MS spectra of IP enriched peptides from (upper) NHP plasma spiked with 10 μg EBOV VLPs protein and (lower) authentic EBOV infected NHP plasma at day 7 post-infection (Animal ID 6311, viral titer $7.91×10^9$ GEq/mL). For simplicity, only the two peptide target peaks were indicated with their mass-to-charge ratio (m/z). (B-C) Internal standard (IS) of EBOV VP40 peptide 8 was spiked into the plasma after digestion for VP40 quantitation. Quantitative standard curves of MS peak intensity ratios between the light (endogenous peptide) and heavy (internal standard peptide) VP40 peptide 8 and the amount of (B) recombinant protein and (C) VLP proteins spiked into NHP plasma. The data is shown as mean with standard deviation (n = 3). Red arrows indicate the assay LOD as 240 or 930 ng/mL in the absence or existence of LDS buffer, respectively.

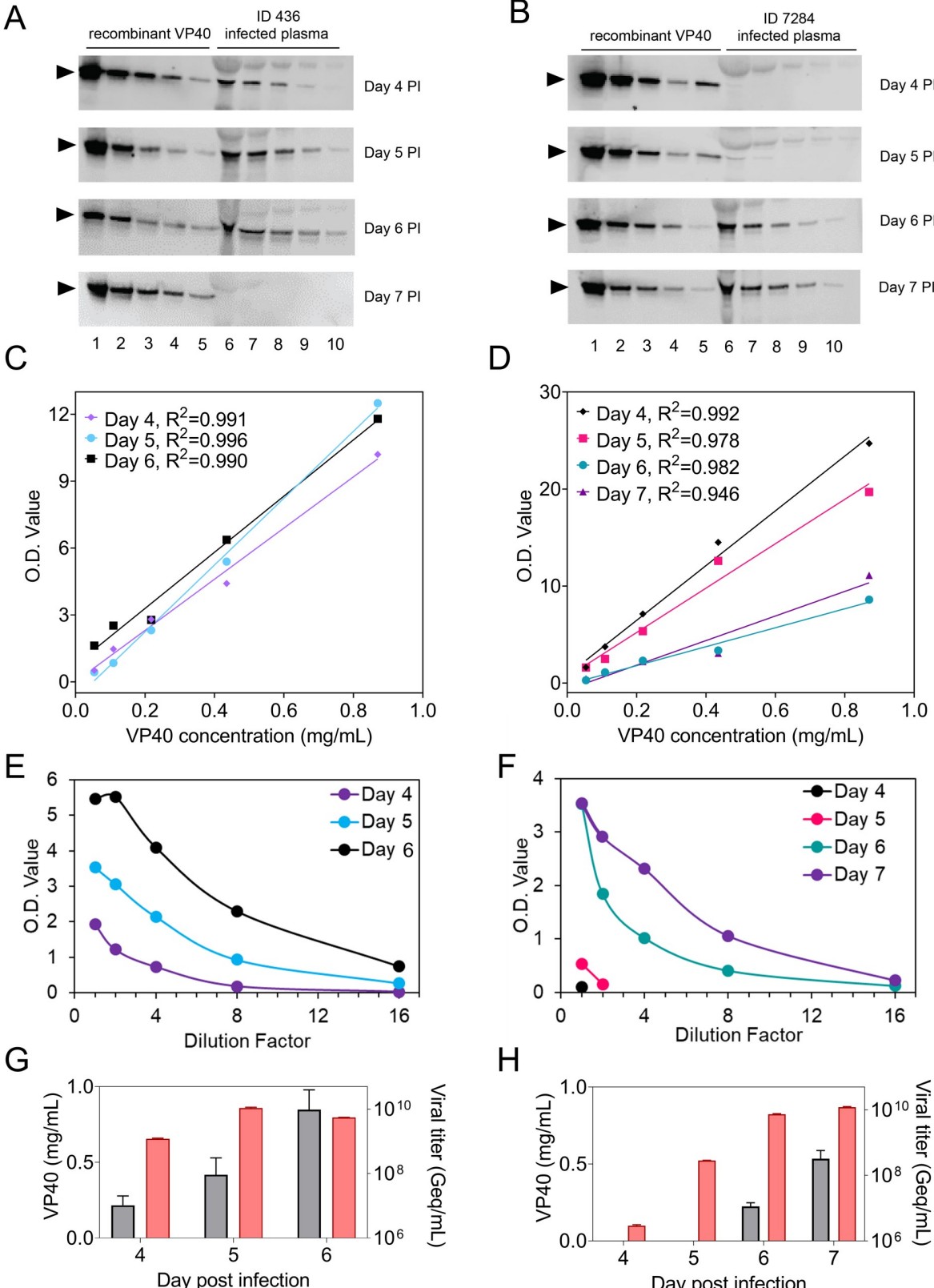

**Fig 7. Western blot analysis of VP40 in NHP plasma with longitudinal collections.** (A-B) A concentration gradient of recombinant VP40 is used as standard, and serial dilution of NHP plasma post-infection (PI) is measured on the same membrane. Well 1–5 represent

for a two-fold decrease of VP40 loading starting from 870 nanogram. Well 6 represents for the undiluted NHP plasma infected with EBOV, and well 7–10 represent for a two-fold serial dilution of the same sample. Two microliters of both diluted and undiluted plasma were loaded into each well. The band of VP40 is indicated by a black arrowhead on each membrane. (C-D) The standard curves of densitometry of VP40 band in each sampling time point for (C) ID 436 and (D) ID 7284 (died at Day 7 PI). The correlation co-efficiency ($R^2$) of each curve is indicated. (E-F) Densitometry values of detected bands in the expected VP40 region in infected plasma dilutions. (G-H) Changes of NHP plasma VP40 concentration (grey bars, left axis) measured by western blot and viral titer (red bars, right axis) measured by RT-PCR during the infection course. The data is shown as mean+SD (n = 3).

LOD of our newly developed IP-MS assay is 2.8 ng/test and 0.7 ng/test in plasma for EBOV and SUDV VP40 respectively when analyzed by MRM, equivalent to $4.6 \times 10^5$ genomes/mL. Ebolavirus viral load showed a wide dynamic range ($10^2$ to $10^8$ Geq/mL) in patients during a 2014 EVD outbreak in Sierra Leone, where mean viral load in serum ranged from $10^5$ to $10^6$ Geq/mL at initial evaluation, typically 6 to 7 days after symptom onset[9]. Therefore, the proposed assay has an acceptable sensitivity for Ebola surveillance.

As a result, the VP40 peptide targets in complex biological samples were confidently identified via tandem MS analysis. Avoiding use of liquid chromatography for peptide separation and ultra-high vacuum turbo pump greatly increases the feasibility of applying this assay in resource-limited area with high prevalence of EVD. The assay LOD of IP-MS analysis of VP40 using MALDImini-1 requires further investigation, specifically to determine the minimal VP40 concentration required to produce high-quality tandem spectra of the two peptide targets.

Another challenge faced by IP-MS detection of VP40 is non-synonymous amino acid variations in the targeted peptide sequence. As reported by Carroll et al., six possible amino acid variations in VP40 protein were observed among 179 genomes from Ebola virus, which were 27 A/V, 123 T/P, 125 F/L, 250 V/I, 269 H/R and 276 V/G[39]. Because the targeted VP40 peptides 8 and 12 locate at position 138–148 and 213–221 respectively, there was no amino acid variation observed in these regions at present. Ebola virus Makona (EBOV-Makona; from the 2013–2016 West Africa outbreak) was a recent isolate of EBOV[40]. The genome of Ebola virus Makona in GenBank with accession ID KJ660347.2, has a VP40 sequence that differs by just two amino acids compared to EBOV. They are 20 A to V and 324 V to I, substitutions which will not affect the IP-MS detection of the two targeted peptides. The targeted regions with single amino acid variation may also be captured by the employed antibodies, as exemplified by the 140 P/Q substitutions between EBOV and SUDV VP40. Currently, VP40 sequence mutations can only be identified by gene sequencing in a reference laboratory. The proposed method could be used for identifying the known VP40 sequence mutations by selecting those peptide regions that cover the mutation sites. Furthermore, our method can be applied to diverse specimen types and inactivation procedures that don't contain LDS or SDS buffer; therefore, it will contribute to further optimization of biosafety practices (e.g., viral inactivation methods) and testing of alternative specimen types such as CSF fluid or seminal fluid for viral persistence in EVD survivors.

In summary, our assay using IP-MS could detect EBOV within 4 hours using a miniature MS instrument. When coupled with LC-MRM, it showed a similar LOD to the antigen-based RDTs that are validated in clinical samples. The ability to differentiate multiple ebolavirus species and quantify VP40 in a species-specific manner make this assay different from the antigen-based RDTs. Public health laboratories that currently use benchtop MALDI instruments for bacterial species identification could use the same instrument for the proposed IP-MS assay of VP40. Further investigation of the feasibility of this method to detect pre-symptomatic EVD patients is needed to employ this assay in a field lab for Ebola surveillance during an outbreak.

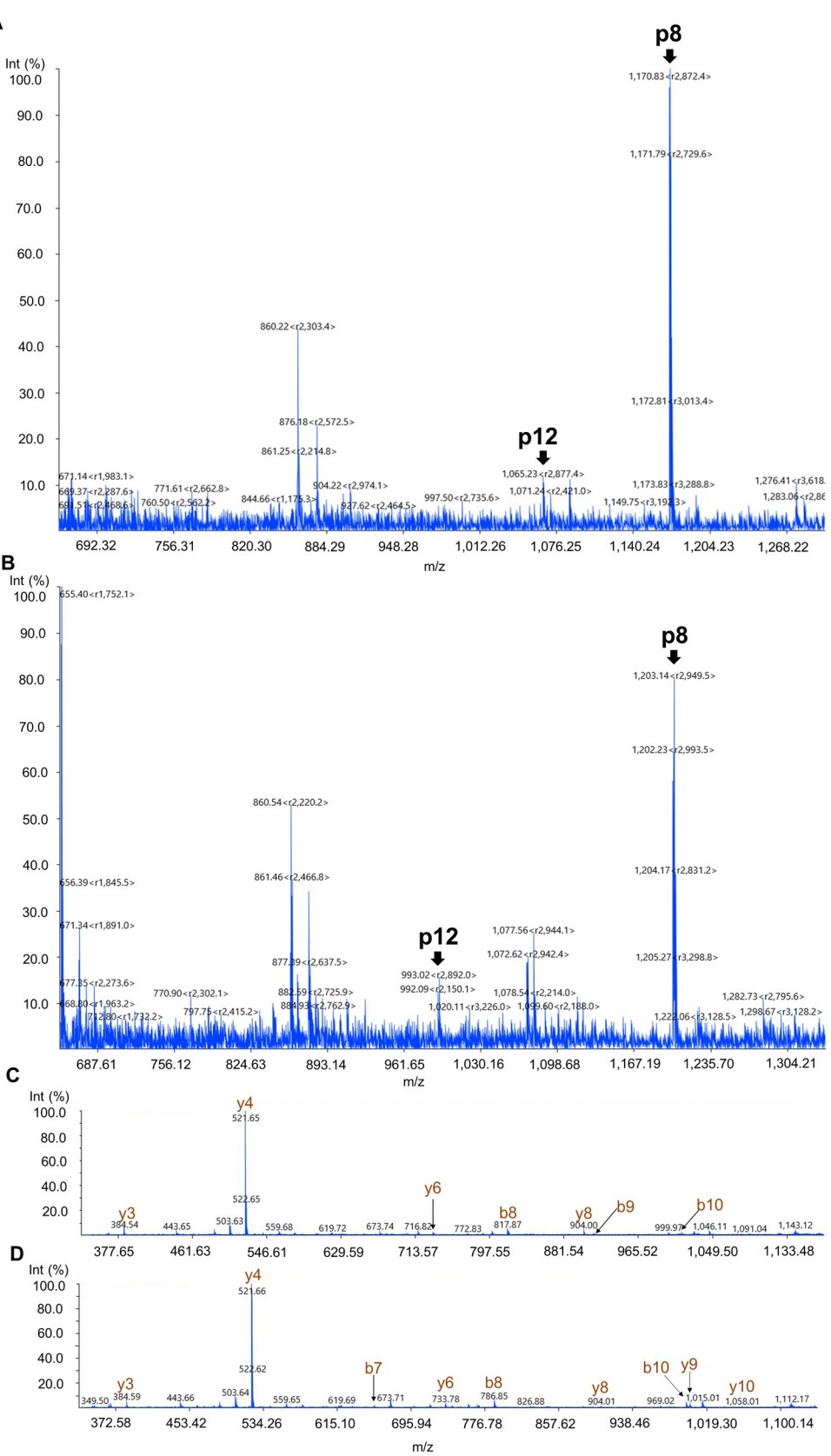

**Fig 8. Detection of VP40 signature peptides in VLP spiked-in samples using miniature MALDI-TOF MS. (A-B)**
Full MS scan of IP enriched peptides from (A) 10 μg/mL EBOV VLP-spiked human plasma sample and (B) from
10 μg/mL SUDV VLP-spiked human plasma sample in the mass range of 660–1,310 m/z. (**C-D**) The tandem MS
spectrum of (C) EBOV VP40 peptide 8 with the sequence LGPGIPDHPLR and (D) SUDV VP40 peptide 8 with the
sequence LGQGIPDHPLR in the mass range of 330–1,140 m/z.

## Methods

### Ethics statement

Non-human primate samples analyzed in this study were collected from archived plasma generated in studies conducted at the Systems and Structural Biology Division, Protein Sciences Branch, U.S. Army Medical Research Institute of Infectious Diseases, which is accredited by the Association for Assessment and Accreditation of Laboratory Animal Care, International and adheres to principles stated in the Guide for the Care and Use of Laboratory Animals, National Research Council, 2011. The parent study that generated these samples was conducted under Institutional Animal Care and Use Committee (IACUC) approved protocols in compliance with the Animal Welfare Act, PHS Policy, and other Federal statutes and regulations relating to animals and experiments involving animals.

### Recombinant VP40, irradiated virions, VLPs and stable-isotope-labeled peptide standard

Recombinant VP40 protein from EBOV and SUDV were obtained from IBT BioServices (Cat. # 0564–001 and 0569–001). Recombinant BDBV VP40 protein was purchased from Sino Biological (Beijing, China; Cat. # 40448-V07E). EBOV and SUDV VLPs were produced under a contract with Paragon Bioservices (Baltimore, MD) using a modification of the procedure described by Warfield et al.[25]. VLPs were manufactured by Paragon Bioservices and were produced by transfecting HEK293F cells with Ebola Zaire virus GP and VP40 genes in pWRG expression vectors, essentially as previously described[41]. Full-length cDNAs encoding ebolavirus (species Zaire, EBOV) VP40 or GP were cloned into the mammalian expression vector pWRG7077. 293T cells were transfected with plasmids using the lipofectamine reagent (Invitrogen, Carlsbad, CA, USA) according to the manufacturer's instructions. Transfected cells were incubated at 37˚C under 5% CO2 for 48 h prior to harvesting VLPs. VLPs were prepared essentially as previously described[42]. Briefly, supernatants were collected 48 h after transfection, overlaid on 20% sucrose and ultracentrifuged at 26 000 rpm for 2 h. Pelleted material was further purified by a sucrose step gradient consisting of 60%, 40% and 10% layers. After ultracentrifugation, the VLPs were recovered at the interface of 10/40% and 40/60% layers. VLP solution protein content was determined by BCA assay (Thermo Pierce; Cat. # 23225). TAFV, RESTV and BDBV irradiated virion solution were obtained from BEI resource (Cat. # NR-44241, NR-44238 and NR-31813). An internal standard peptide LGPGIPDHPLR that contained a stable-isotope-labeled arginine ($^{13}C_6$$^{15}N4$) was synthesized by Thermo Scientific. A water solution containing 500 nM internal standard was prepared.

### Sequence alignment of VP40

Six VP40 protein sequences [Q5XX06 (SUDV), Q05128 (EBOV), Q8JPX9 (RESTV), B8XCM9 (BDBV), B8XCN8 (TAFV) and A0A343EQF9 (BOMV)] were retrieved from the UniProt website (www.uniprot.org), and aligned using the Clustal Omega multiple sequence alignment function that is embedded in the UniProt website.

## Generation of VP40 peptide antibodies

A custom mouse monoclonal antibody specific for the EBOV VP40 peptide LGPGIPDHPLR and rabbit polyclonal antibodies specific for the BDBV VP40 peptide LRPILLPGK were generated for this study by GenScript (Nanjing, China), using synthetic peptides conjugated to the carrier protein keyhole limpet hemocyanin (KLH) as the immunogen. The mouse and rabbit polyclonal antibodies were purified from culture medium and serum, respectively, by protein A chromatography, after which their concentration was determined by BCA assay. For the final monoclonal antibody screening step, antibodies purified from the culture media of each subclone were analyzed by IP-MS to evaluate their relative ability to detect VP40 target peptide in human plasma spiked with 2 μg of EBOV VLP sample after trypsin digestion.

## Non-human primate EBOV infection and plasma collection

Adult rhesus macaques were intramuscularly inoculated with a target titer of 1000 plaque-forming units (PFU) of EBOV (H.sapiens-tc/COD/1995/Kikwit-9510621) and the infection course and sample collection were performed as previously described[10]. These animals served as control animals in previously executed therapeutic studies, and the samples were retrospectively analyzed, the animals' sex and age information are available in **S3 Table**. All EBOV studies were conducted in Animal Biosafety Level 4 containment. Plasma samples from these studies were inactivated by heating to 95˚C for 10 min after the addition of a 1:3 volume of 4× NuPAGE LDS Sample Buffer and Sample Reducing Agent (Invitrogen; Cat. # NP0007 and NP0009), after which samples were stored at -80˚C until use.

## Filter-aided sample preparation (FASP) for infected NHP plasma

Plasma samples (6 time points/animal) were first processed in BSL-3 or BSL-4 containment by adding 25 μL SDS-PAGE solubilizing/reducing buffer to 75 μL sample and heating to 95˚C for 10 min. Samples were then removed from containment and stored at − 80˚C until processed by the iFASP method. Briefly, for each sample, five aliquots each containing 10 μl of inactivated plasma were processed in parallel. Each aliquot was diluted with 200 μl of 8M Urea/0.1M HCl pH 8.5 (UT8). The diluted sample was added to a filter assembly (Microcon-30kDa centrifugal filter unit with ultracel-30 membrane Millipore MRCF0R030) and centrifuge at 14,000 ×g for 25 minutes. The flow-through was discarded, and 200 μl of UT8 followed by 100 μl of 55mM iodoacetamide in UT8 was added to the filter assembly and spin at 14,000 ×g for 25 minutes. The samples were incubated in the dark, while shaking, for 25 minutes at room temperature. Centrifuge the filter assembly tubes at 14,000 ×g for 15 minutes. Discard the flow-through, add 100 μl of UT8. Centrifuge at 14,000 ×g for 15 minutes. Discard the flow-through. Repeat the washing step with 100 μl of UT8 two more times. Add 100 μl of trypsin solution (0.04ug/ul in 25 mM Tris-HCl pH 8). Cap the filter unit and tightly wrap with parafilm to prevent evaporation. Incubate at 37˚C, while shaking, for 16 hours. Centrifuge at 14,000 ×g for 15 minutes. Add 40 μl of 25 mM Tris-HCl pH 8. Centrifuge at 14,000 ×g for 15 minutes. Repeat the elution step with 40 μl of 25 mM Tris-HCl one time. Add 50 μl of 0.5 M NaCl. Centrifuge at 14,000 ×g for 15 minutes. Combine the eluent from all five aliquots and adjust pH to 8 with 1 M tris buffer.

## Western blot

Recombinant EBOV VP40 was serially diluted (2-fold) to generate a 5-point standard curve (0.87μg-0.054ug). These standard dilutions were size-fractionated under reducing conditions on a 4–12% NuPAGE bis-tris polyacrylamide gel (Thermo Fisher Scientific; Cat. # NP0321)

adjacent to 2-fold serial dilutions of EBOV VLP samples or plasma samples of EBOV-infected NHPs for relative quantification of VP40 content in these samples. VLP samples (2.5µg total protein in 1µL) were size fractionated in four adjacent lanes (undiluted to 1:8 dilution), while NHP plasma samples were loaded in four or five adjacent lanes (2 µL neat plasma to 1:8 or 1:16 dilution). After transfer to PVDF, blots were blocked overnight with Pierce Protein-Free Blocking Buffer (Thermo Fisher Scientific; Cat. # 37572) and then incubated overnight at 4˚C on a rocking platform with a 1:500 dilution of a mouse monoclonal VP40 antibody (IBT Bio-Services; Cat. # 0201–016), washed 3× with PBS and 0.1% Tween-20 (PBST) for 5 min each, and then hybridized for 1 hr at 25˚C in PBST with a 1:10,000 dilution of a goat anti-mouse IRDye 680-labelled antibody (LICOR Biosciences; Cat. # 926–68070). Blots were then washed 3× washed with PBST and stored in PBS until visualized using an Odyssey infrared imaging system (LI-COR Biosciences; model # 9210). VP40-specific densitometry values obtained for the VLP and NHP plasma samples were plotted against those of the recombinant standard curve for the corresponding blot to extrapolate the VP40 content of the VLPs and plasma samples, and VLP VP40 concentrations were determined by averaging all extrapolated values from two replicate gels.

## IP-MS analysis of VP40 in human plasma

VP40 peptide-specific antibodies generated for this study were diluted in 1× pH 7.4 PBS (Fisher Scientific, Cat. # BP399-4) and 50 µg antibody was conjugated with 3mg of protein G-coupled Dynabeads (Thermo Fisher Scientific; Cat. # 1009D) in 400 µL coupling buffer (PBS pH 7.4 with 0.2% (v/v) Tween-20, Sigma; Cat. # P9416-100 ML) according to the manufacturer's instructions. Beads were washed with 200 µL coupling buffer once before antibody conjugation, twice following conjugation, and then resuspended in 1 mL coupling buffer.

Plasma from a single healthy human donor (Valley Biomedical, VA) was split into 50 µL aliquots that were spiked to contain 3.125–50 nM of recombinant SUDV, EBOV or BDBV VP40 protein, or spiked with 1 µL of a SUDV or EBOV VLP sequential dilutions containing 2.5 µg–2.5 ng total protein. VP40-spiked and VLP-spiked human plasma, and inactivated plasma from EBOV-infected non-human primates were mixed with 500 µL of 0.5% (v/v) Triton X-100 (Sigma; Cat. # X100-500ML), incubated for 5 min in a 100˚C heat block, cooled for 5 min in a 25˚C water bath, then mixed with 10 µL of an unbuffered 1 M Tris base solution (Bio-Rad; Cat. #161–0716) to attain a pH 8.5 solution. Samples were then spiked with 10 µg of sequencing grade modified trypsin, digested at 37˚C for 1 hr with rotary mixing, and then spiked with 5 µL of 10% (v/v) trifluoracetic acid (TFA) to achieve pH 7. After digestion, samples to be analyzed for target peptide quantification were spiked with 20 µL of a 5 nM stable-isotope-labeled internal standard peptide solution. Digested plasma samples were then incubated with antibody-conjugated protein G Dynabeads (0.15 mg beads per antibody per sample) for 1 hr at 25˚C with rotary mixing for target peptide capture. The captured peptides were eluted and analysis by MS.

## MS analysis of tryptic peptides of recombinant EBOV and SUDV VP40

Recombinant EBOV and SUDV VP40 (5 µg each; IBT BioServices, Rockville, MD) were dissolved in 100 µL of 50mM ammonium bicarbonate, then mixed with 0.5 µg of trypsin sequencing grade modified trypsin (Promega; Cat. # V5111), and incubated at 37˚C for 16 hrs. After digestion, 1 µL was spotted on a MALDI target plate, followed by 1 µL of matrix solution containing 4 mg/mL α-Cyano-4-hydroxycinnamic acid (CHCA, Sigma; Cat. # 70990-1G-F), 50% (v/v) acetonitrile, 0.1% (v/v) trifluoracetic acid (TFA) and 49.9% (v/v) water. Mass spectra in the range of 500–3,000 were then collected on a Microflex LRF MALDI-TOF MS system

(Bruker Daltonics Inc., Billerica, MA), using 3000 shots collected at 70% laser power. A 5 µL aliquot was also loaded onto an Orbitrap Fusion Lumos mass spectrometer (Thermo Fisher Scientific) coupled with an UltiMate 3000 ultrahigh-pressure liquid chromatography (UHPLC) system. Peptides were loaded onto an Acclaim PepMap100 C18 trap column (300 µm ID × 5 mm, 5 µm, Thermo Fisher Scientific; Cat. # 160454) and then separated on a PepMap RSLC C18 analytical column (75 µm ID×15 cm, 3 µm, Thermo Fisher Scientific, Cat. # 164568). Peptides were then eluted with a 300 nL/min gradient generated by mixing buffer A (0.1% formic acid in water) and buffer B (0.1% formic acid, 80% acetonitrile in water) as follows: a 5 min wash with 5% buffer B, a 17 min 5–38% buffer B gradient, a 2 min 38–95% buffer B gradient, a 0.1 min 95–5% buffer gradient, and a 1 min 5% buffer B wash. The resolutions of the survey scan and tandem mass scans were 120,000 and 30,000 respectively, across a mass-to-charge ratio window of 350–1,700, and the tandem mass scan was collected in data dependent mode with a fixed cycle time of 3 s.

## MS analysis of immunoprecipitated VP40 peptides

Peptide-hybridized beads were transferred and washed two times with 100 µL of PBS, and once with 100 µL of LC grade water. Beads were then incubated for 30 min at 25°C with 100 µL of 1% (v/v) formic acid solution followed by magnetic separation to obtain supernatants containing eluted peptides. StageTips[43] were made by packing 200 µL pipette tips with four layers of Empore C8 solid phase extraction disk (3M; Cat. # 2214-C8), washed by 25°C centrifugation at 1000g for 3 min with 50 µL of 0.1% (v/v) TFA acetonitrile solution and 50 µL of 0.1% (v/v) TFA. Each peptide sample was split and loaded onto two StageTips, and captured peptides were eluted with 50 µL of 0.1% (v/v) TFA acetonitrile solution. Paired eluents were combined, dried by vacuum concentration, re-dissolved with 8 µL of sampling buffer containing 0.1% (v/v) formic acid, 2% (v/v) acetonitrile, and centrifuged at 25°C and 21,000 g, for 10 min.

MALDI-TOF MS analysis was conducted as described above. MRM-MS analysis of captured peptides was performed using an EASY-nLC 1000 HPLC system coupled with an Altis triple quadrupole mass spectrometer (Thermo Scientific), using the same column and LC conditions used for the original LC-MS analysis of the recombinant VP40 tryptic peptides. Both Q1 and Q3 resolution (full width at half maximum, FWHM) were set to 1.2. These analyses used a 1.5 mTorr collision gas pressure, a 2.2 kV spray voltage, a 275°C capillary temperature, using a 40 V collision energy setting and a 40 msec dwell time for each transition, with a 131 V RF lens setting.

Miniature MS analysis of captured peptides was performed using a MALDImini-1 Mass Spectrometer (Shimadzu, US). All samples were spotted with 5mg/ml CHCA. The samples were spotted in a 1:1 volume with CHCA matrix on the MALDI target.

## Data analysis

Tandem mass spectra were searched against EBOV and SUDV VP40 protein sequences to identify corresponding peptides using Mascot Server software (version 2.6, Matrix Science, Boston, MA). MRM-MS data were imported into FreeStyle software (version 1.5, Thermo Fisher Scientific), and the signal-to-noise ratio (SN) of peptide peaks in these extracted ion chromatograms were calculated using the Genesis peak detection algorithm integrated into this software, with parameters set to detect all peaks with a SN $\geq$2. The identified spectra of peptide targets from EBOV and SUDV were imported into the Skyline software (version 20.1.0.76, MacCoss Lab, University of Washington) to generate a spectral library. The similarity score between query spectrum and library spectrum (dotp, for both p12 and p8), and

between query spectrum and its internal standard (rdotp, for p8 only), as well as peak area of each targeted peptide was exported, and a standard curve was built by normalizing the peak area of a target peptide produced by the spiked VLPs to that of a stable-isotope-labeled internal standard peptide. The assay LOD was estimated based on two criteria: (1) the two peptides had a dotp ≥ 0.8 considering both types of MS instruments used to generate library spectra and query spectra; (2) the peak area ratio of p8 was significantly different from that found in the next lower diluted standard. Linear standard curves and their correlation coefficients were generated using GraphPad Prism (version 8.4.1, GraphPad Software). All data for LOD estimation and standard curve generation were provided in **S2 Table**. The spectra collected by the MALDImini-1 MS was analyzed by the eMSTAT Solution software (Shimadzu, US). *De novo* peptide sequencing was performed *in silico* for all fragmented peptides using ProteinProspector (UCSF, US). The MS data obtained from Bruker MALDI-TOF MS systems were processed using mMass (http://www.mmass.org/), after converting the raw data into mzML format. Peak picking was conducted by mMass with the SN cutoff as 1, and all other parameters as default settings. The two peptide peaks were manually annotated and their SNs were exported.

## Supporting information

**S1 Fig. The species-specificity of three VP40 tryptic peptides and their masses distribution among ebolavirus species.**
(TIF)

**S2 Fig.** MRM analysis of two VP40 peptide variants from the PBS solution spiked with 0.2 μg/μL total viral protein of inactivated authentic virions from (A) RESTV, (B) BDBV and (C) TAFV.
(TIF)

**S3 Fig.** Reproducibility of extracted peak area of (A) VP40 peptide 8 with sequence LGPGIPDHPLR shared by the three species, (B) VP40 peptide 12 with sequence LRPILLPGK shared by BDBV and RESTV, and (C) VP40 peptide 12 with sequence LRPILLPGR from TAFV.
(TIF)

**S4 Fig. VP40 peptides identified by MALDI-TOF MS analysis of recombinant protein digest.**
(TIF)

**S5 Fig. MALDI-TOF MS spectra of IP enriched peptides from human plasma spiked with 12.5 nM recombinant VP40.**
(TIF)

**S6 Fig. Linear correlation between VP40 content determined by western blot and viral titer determined by RT-PCR.**
(TIF)

**S1 Table. VLP VP40 mass content determined by MRM analysis of p12**
(XLSX)

**S2 Table. Multiplex MRM analysis of p8 and p12 in VLP-spiked human plasma for standard curve generation.**
(XLSX)

**S3 Table. Viral load and demographic data for the NHP models.**
(XLSX)

**S1 Data. IP-MALDI-MS assay LOD in plasma spiked with recombinant VP40.**
(DOCX)

## Acknowledgments

We thank Dr. Jordan Witkop and Ryan Walsh (Shimadzu) for their help in collecting the MS spectra of peptides using the MALDImini-1. The following reagent was obtained through BEI Resources, NIAID, NIH: Taï Forest Ebolavirus, Ivory Coast, Gamma-Irradiated, NR-44241; Bundibugyo Ebolavirus, Prototype Isolate #811250 (200706291 Uganda), Gamma-Irradiated, NR-31813; Reston Ebolavirus, 119810 RIID, Gamma-Irradiated, NR-44238.

## Author Contributions

**Conceptualization:** Qingbo Shu, Jia Fan, Lisa H. Cazares, Tony Y. Hu.

**Data curation:** Qingbo Shu, Tara Kenny.

**Formal analysis:** Qingbo Shu, Tara Kenny, Christopher J. Lyon.

**Funding acquisition:** Lisa H. Cazares, Tony Y. Hu.

**Investigation:** Lisa H. Cazares, Tony Y. Hu.

**Methodology:** Qingbo Shu, Tara Kenny, Lisa H. Cazares, Tony Y. Hu.

**Project administration:** Jia Fan, Lisa H. Cazares.

**Resources:** Jia Fan, Lisa H. Cazares, Tony Y. Hu.

**Supervision:** Lisa H. Cazares, Tony Y. Hu.

**Validation:** Tara Kenny, Lisa H. Cazares, Tony Y. Hu.

**Visualization:** Qingbo Shu, Christopher J. Lyon.

**Writing – original draft:** Qingbo Shu, Christopher J. Lyon, Lisa H. Cazares.

**Writing – review & editing:** Qingbo Shu, Jia Fan, Christopher J. Lyon, Lisa H. Cazares, Tony Y. Hu.

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
