## [Decision Letter · Decision Letter 0]

19 Jul 2021

Dear Dr Hu,

Thank you very much for submitting your manuscript "Species-specific Quantification of Circulating Ebolavirus Burden using VP40-derived Peptide Variants" for consideration at PLOS Pathogens. As with all papers reviewed by the journal, your manuscript was reviewed by members of the editorial board and by several independent reviewers. In light of the reviews (below this email), we would like to invite the resubmission of a significantly-revised version that takes into account the reviewers' comments.

Both reviewers found this to be an interesting and informative manuscript with potential for describing a new diagnostic tool for distinguishing ebolavirus species. The reviewers make several suggestions for improvement including a better correlation across the ebolavirus species and a discussion of the advantages of this method over standard diagnostics.

We cannot make any decision about publication until we have seen the revised manuscript and your response to the reviewers' comments. Your revised manuscript is also likely to be sent to reviewers for further evaluation.

Sincerely,

Amy L Hartman, PhD

Associate Editor

PLOS Pathogens

Christopher Basler

Section Editor

PLOS Pathogens

Kasturi Haldar

Editor-in-Chief

PLOS Pathogens

orcid.org/0000-0001-5065-158X

Michael Malim

Editor-in-Chief

PLOS Pathogens

orcid.org/0000-0002-7699-2064

Both reviewers found this to be an interesting and informative manuscript with potential for describing a new diagnostic tool for distinguishing ebolavirus species. The reviewers make several suggestions for improvement including a better correlation across the ebolavirus species and a discussion of the advantages of this method over standard diagnostics.

Reviewer's Responses to Questions

**Part I - Summary**

Reviewer #1: In the manuscript entitled “Species-specific quantification of circulating Ebolavirus burden using VP40-derived peptide variants,” Shu and colleagues develop an immunoprecipitation (IP)-based mass spectrometry (MS) assay to distinguish human-pathogenic ebolaviruses based on amino acid differences in VP40. The authors first identify two immunogenic tryptic peptides that, together, can differentiate between EBOV, SUDV, BDBV, and TAFV based on their amino acid sequences. The authors then use these peptides to produce antibodies, and they incorporate these antibodies into their IP-MS assay. Using this assay the authors were able to distinguish between EBOV, SUDV, and BDBV VP40 in spiked human plasma. The authors were also able to detect EBOV in samples obtained from infected NHPs. Together, these data suggest that this novel IP-MS assay could be used as a diagnostic tool, although its advantages in comparison to standard RT-qPCR diagnostic tests are not clear. Indeed, while this study provides proof of principle, additional work is required to demonstrate whether the IP-MS assay described here will provide any use in a clinical or field setting.

Reviewer #2: In “Species-specific Quantification of Circulating Ebolavirus Burden using VP40-derived Peptide Variants”, Shu et al. describe the development of a quantitative assay for EBV using species-specific VP40 peptide variants. The objectives of the study are clearly defined and articulated, with a strong experimental method, excellent figures, and a well written results section. The discussion goes some way to listing the strengths and weaknesses of the data, the latter of which is the absence of any clinical material validation using the methods described. However, the assays parameters are discussed in relation a comparable method (the LFI ReEBOV RDT) and other more standard diagnostic tests (qRT-PCRs) are included which offers the reader an idea of where the newly described technique would stand in a diagnostic setting.

The data presented is very interesting, and shows the potential of Ag-based diagnostics in for these pathogens. I also appreciate the way the authors included thought as to how these assay could be adapted to a field setting which is important for pathogens such as EBV. The main comments I have on the data are that: a) the absence of any data for the remaining EBV species; and b) it is lacking any human clinical samples which would greatly strengthen the data here (although I understand that EBV clinical material is not the easiest to obtain, and the NHP plasma samples represent a suitable surrogate for this).

**Part II – Major Issues: Key Experiments Required for Acceptance**

Reviewer #1: Can the authors provide the details for the RT-PCR assay that was used to obtain the data shown in Fig. 5G, H? Were the primers/probe used in this assay validated for diagnostic use? The results in Fig. 5 indicate that the RT-PCR assay is more sensitive than the IP-MS assay, which would make the latter assay less useful as a diagnostic test. Can the authors comment on this discrepancy?

The authors position their IP-MS assay as a potential alternative diagnostic test to the standard RT-PCR test. However, based on the description of the IP-MS assay in the Results and Materials and Methods sections, it is not clear whether the IP-MS assay improves upon the “weaknesses” that the authors list on Lines 58-59 for RT-PCR (i.e., cost, time, availability, and operator expertise). Likewise, it is not clear whether the IP-MS is rapid or simple (Line 61). Based on the protocols outlined in the Materials and Methods section, it seems that numerous steps over several hours to days are required to perform the IP-MS, which would make it significantly more complicated than the standard RT-PCR test. Can the authors clarify whether their IP-MS assay is truly a rapid, simple, and inexpensive diagnostic test this, especially in relation to RT-PCR. It would also be helpful to understand how practical the IP-MS assay would be to perform in a resource-limited setting. One suggestion would be to include a flow chart that outlines the steps for IP-MS and RT-PCR tests from clinical sample to results.

Can the authors evaluate their IP-MS assay using BDBV and TAFV VLPs as well? Without these data it is difficult to conclude that the assay is capable of differentiating between all human-pathogenic ebolaviruses. Further, if at all possible, the authors should consider evaluating additional samples from infected NHPs (or other animal models) infected with EBOV, SUDV, or BDBV. Additional data points would be extremely helpful in demonstrating that this assay is robust and specific.

Reviewer #2: • Given that the nucleotide sequences are published, and the system exist to generate the VLPs (used here), I think data where BDBV and TAFV VLPs were analyzed should be included. I know that some BDBV is included in the supplementary material, but given as it is mentioned in the abstract and introduction, I think it should be included in the main text. However, if TAFV (and potentially RESV and BOMV?) were excluded from the analysis early on (presumably due to lack of fatal disease in humans) then this should be clearly articulated in a sentence early on to avoid confusion.

• Given that the authors describe the potential diagnostic uses of this technology (which is probably the most novel and exciting aspect of this work) I think more work needs to be carried out in this area. As previously mentioned, human clinical samples would be the idea choice here, but given the difficulty of obtaining these (with detailed matched clinical details) NHP plasma represent a suitable and ideal surrogate. However, the data would be much stronger if greater numbers were analyzed here rather than the 2 animals currently included. The one important question that I think could be answered here to increase manuscript strength would be “how early in infection can VP40 be detected”. This data, along with its variability (this is where I think having more numbers would be very useful) would be extremely useful and, if possible, should be included somewhere in the manuscript.

**Part III – Minor Issues: Editorial and Data Presentation Modifications**

Reviewer #1: Lines 13-14: Reston virus is not likely a human pathogen, whereas Tai Forest virus has caused at least one recorded case of severe disease in a human.

Line 15: Spelling error: “caused”

Line 25, 44-45: Filovirus nomenclature in this line should be correct. “EBOV” and “SUDV” refer to the viruses, not the species. Species cannot cause lethal disease, but viruses can.

Line 26 and elsewhere: As far as this reviewer is aware, “EBV” is not a commonly used or accepted abbreviation for “ebolavirus.” Consider replacing all instances of “EBV” with “ebolavirus.”

Line 26: The authors state that the proposed assay differentiates four of the six known ebolavirus species, but the manuscript primarily describes assays performed with material derived from EBOV and SUDV, with some assays including material derived from BDBV. Although the proposed IP-MS assay theoretically allows differentiation of TAFV, the authors do not provide these data. Suggest revising this statement accordingly.

Line 157: Based on Fig 1E, it appears that VP40 p8 is conserved among all human-pathogenic ebolaviruses except for SUDV, not EBOV.

Line 271: Spelling error: “BDBV”

Line 284: Spelling error: “authentic”

Line 323: Were plasma samples obtained from NHP 436 on Day 7 PI? If so, why was no VP40 signal detected in the Western blot in Fig. 5A?

Lines 353-357: It is a bit misleading to begin the discussion section with a description of virus isolation as the gold-standard method for confirming ebolavirus infection. While this is true, it does not seem likely that the IP-MS assay will be able to compete with this gold-standard. Rather, the IP-MS assay is, at best, an alternative to RT-PCR assays and, perhaps, RDT assays.

Lines 383-385: Suggesting re-writing this sentence to avoid confusion.

Lines 407-409: The authors seem to suggest that their IP-MS assay could be used to identify VP40 mutations. While this may be true specifically for the single mutation in p8 and the three mutations in p12 that are assessed in this study, it would not necessarily be true for other mutations within these peptides, would it? In any case, it does not seem practical to suggest that IP-MS be used to define protein mutations when gene sequencing can be performed easily, including in the field.

Reviewer #2: • Line 44: Expand EBV

• Line 157: Should be SUDV rather than EBOV I think?

• Line 207: glycoprotein should not be capitalized

• Line 321: Include the type of serial dilution (2-fold I think?) here.

• Line 359-362: Line discussing the types of inactivation should have a reference (or multiple references).

• Line 431: Given their importance in this work, in this section I think the specifics of how the VLPs were generated, purified, and quantified should be expended on rather than refereeing to a single reference.

• Line 464: Age, sex, and route of inoculation for the NHPs should be included.

• Line 533: Missing a VP40 after the SUDV I think?

• Figure 1D: No reference in the legend to what the blue and purple (closed an open) dot represent, or what the dotted blue line represents.

• Figure 3A: Several artifacts present, looks like from masking boxes. Also, this makes it look as if the VP40 and GP genes are expressed as a single CDS? If the whole plasmid is represented here then each (relevant) CDS should be individually displayed. It would be useful to show what the arrows represent (presumably transfection and VLP release) as well.

• Figure 3B: What do the wells represent? If they are dilution of the VP40 then this should be indicated on the figure. A protein weight marker would also be relevant here.

• Figure 4C: this is not referred to in the main text.

• Figure 5A and B: Again, please indicate what each of the wells contains, and include a protein size marker if possible.

• Figure 5G and H: Error bars for the RT-PCR data (red bars)?

• Supporting info S6 data: should be BDBV in the title?

PLOS authors have the option to publish the peer review history of their article (what does this mean?). If published, this will include your full peer review and any attached files.

Reviewer #1: No

Reviewer #2: No
---

## [Editor Report · Decision Letter 1]

14 Oct 2021

Dear Dr Hu,

We are pleased to inform you that your manuscript 'Species-specific Quantification of Circulating Ebolavirus Burden using VP40-derived Peptide Variants' has been provisionally accepted for publication in PLOS Pathogens.

Best regards,

Amy L Hartman, PhD

Associate Editor

PLOS Pathogens

Christopher Basler

Section Editor

PLOS Pathogens

Kasturi Haldar

Editor-in-Chief

PLOS Pathogens

orcid.org/0000-0001-5065-158X

Michael Malim

Editor-in-Chief

PLOS Pathogens

orcid.org/0000-0002-7699-2064

Thank you for thoroughly addressing the reviewers concerns and resubmitting a revised manuscript. The flow chart in revised Fig. 2 and the addition of more filovirus species in Fig. 5 enhance the impact of the overall manuscript.
---

## [Editor Report · Acceptance letter]

3 Nov 2021

Dear Dr Hu,

We are delighted to inform you that your manuscript, "Species-specific Quantification of Circulating Ebolavirus Burden using VP40-derived Peptide Variants," has been formally accepted for publication in PLOS Pathogens.

Best regards,

Kasturi Haldar

Editor-in-Chief

PLOS Pathogens

orcid.org/0000-0001-5065-158X

Michael Malim

Editor-in-Chief

PLOS Pathogens

orcid.org/0000-0002-7699-2064